# Development of the Aerial Remote Triage System using drones in mass casualty scenarios: A survey of international experts

**Cristina Álvarez-García**[1], **Sixto Cámara-Anguita**[2], **José María López-Hens**[2], **Nani Granero-Moya**[3]*, **María Dolores López-Franco**[1]*, **Inés María-Comino-Sanz**[1], **Sebastián Sanz-Martos**[1], **Pedro Luis Pancorbo-Hidalgo**[1]

**1** Department of Nursing, University of Jaén, Jaén, Spain, **2** Emergency Medical Drone Co-operative, Jaén, Spain, **3** Basic Health Area Baeza, District Jaén nordeste, Servicio Andaluz de Salud, Jaén, Spain

* nanigranero@gmail.com (NGM); mlfranco@ujaen.es (MDLF)

**Citation:** Álvarez-García C, Cámara-Anguita S, López-Hens JM, Granero-Moya N, López-Franco MD, María-Comino-Sanz I, et al. (2021) Development of the Aerial Remote Triage System using drones in mass casualty scenarios: A survey of international experts. PLoS ONE 16(5): e0242947. https://doi.org/10.1371/journal. pone.0242947

## Abstract

The use of drones for triage in mass-casualty incidents has recently emerged as a promising technology. However, there is no triage system specifically adapted to a remote usage. Our study aimed to develop a remote triage procedure using drones. The research was performed in three stages: literature review, the development of a remote triage algorithm using drones and evaluation of the algorithm by experts. Qualitative synthesis and the calculation of content validity ratios were done to achieve the Aerial Remote Triage System. This algorithm assesses (in this order): major bleeding, walking, consciousness and signs of life; and then classify the injured people into several priority categories: priority 1 (red), priority 2 (yellow), priority 3 (green) and priority * (violet). It includes the possibility to indicate save-living interventions to injured people and bystanders, like the compression of bleeding injuries or the adoption of the recovery position. The Aerial Remote Triage System may be a useful way to perform triage by drone in complex emergencies when it is difficult to access to the scene due to physical, chemical or biological risks.

## Introduction

Nowadays, the new technologies are highly important in our society, so their use is increasing in the healthcare field [1]. These technologies help speed up the delivery of assistance in some life-threatening situations [2]. Recently, the use of unmanned aerial vehicles or drones in health emergencies has increased. One of the main benefits of the use of drones is that they avoid endangering rescuers [3–6] in cases of shootings, fires, radiation or the presence of infectious agents, explosives, smoke or gases [7,8]. Drones are widely used in health emergencies because they can cover large distances in a short time and access places where rescuers had trouble reaching [9,10], for example, rural environments [11].

By using a drone, it is possible to evaluate several important factors in a mass-casualty incident (MCI), such as the type of incident, magnitude, additional hazards, number and location of injured people and possible access and evacuation routes [8,12–14]. The use of drones is

**Data Availability Statement:** All files are available from the figshare database (https://doi.org/10.6084/m9.figshare.14199272.v2).

**Funding:** The authors received no specific funding for this work.

**Competing interests:** SCA and JMLH are founders of Emergency Medical Drone cooperative. The authors have no conflict of interest to declare. We do not consider any opposed reviewers. This does not alter our adherence to PLOS ONE policies on sharing data and materials.

accurate and safe in the identification of hazards in MCI scenarios [15]. Early evaluation of the incident area will facilitate earlier establishment of an action plan and improve the management of the MCI [16]. However, it is necessary to consider the recommendation of the Committee on Guidance for Establishing Crisis Standards of Care for Use in Disaster Situations, there is an obligation to make a prudent assessment in the allocation of available resources and to avoid situations where healthcare resources are diverted from the emergency to the deployment of drones without well defined, reasonable and informed objectives [17].

Recently the use of drones for triage in MCIs has been proposed, with the START (Simple Treatment and Rapid Transport) triage method [4]. That study found the same accuracy for triage when it was performed with a drone or in a standard manner, although the triage with a drone was about three and a half minutes slower. However, we cannot forget that in rural areas a drone can arrive 93% faster than an emergency medical service (EMS), so it allows a remote triage before arriving at the scene of the accident. In Spain, drones have been used for an assessment of the scene before the arrival of the EMS, and it was found that on-site triage using the START method after the drone assessment could correctly identify more injured people (92%) that a triage without previous drone inspection (62%) [18]. Another study used a drone for triage with the SALT method and to organize resources in a simulated terrorist attack with explosives, found that 82% of the participants adequately classified 12 of the 15 injured people [16].

There are several triage systems for MCIs, including START (Simple Treatment and Rapid Transport), Triage Sieve, Care Flight Triage, STM (Sacco Treatment Method), Homebush, Military Triage, CESIRA, SALT (Sort-Assess-Lifesaving Interventions-Treatment/Transport) [19]. All triage systems classify injured people by a color code, taking into account physiological parameters such as the ability to walk, breathe, perfusion/pulse or response to commands. These codes are 1- deceased or expectant (black), 2- immediate attention (red), 3- delayed attention (yellow), or 4- ambulatory (green). Some systems include the code white to classify suspected dead or blue for contaminated injured people [20,21].

However, none of the existing systems are applicable in all situations, since their use is not specified, for example in situations where there may be biological, chemical, radiological or infection risks [21], poor illumination [18], rural, or mountainous areas, in short in MCIs or emergencies that are complex or difficult to access [22]. So, the objective of our study was the development and initial testing of a remote triage procedure using drones.

## Materials and methods

This study had three phases: 1- Literature review, 2- Development of the remote triage algorithm, and 3- Evaluation of the concept and the algorithm by an expert panel.

### Phase 1. Literature review

The search for published studies was performed in online databases: Global Health, Web of Science, Scopus, Cochrane, CINAHL, Health and Medical, Medline, and ScienceDirect; and also in Cuiden Plus, Dialnet Plus, IME and LILACS for the Iberoamerican literature. The search strategy used the terms earthquake, emergency medicine, drones, terrorist attacks, triage and unmanned aerial vehicle, in English and Spanish, when appropriate, retrieving articles published from inception until January 2021. A reverse search was also carried out from the reference list of each one of the initially retrieved articles looking for additional articles in the field.

## Phase 2. Development of a remote triage algorithm using drones

The first version of the remote triage algorithm was developed from the literature review, based on existing triage systems and practical experience of some of the authors. This first version covered the following aspects: 1-mayor bleeding, 2- walking, 3- consciousness/alertness, 4- signs of life. This version classified the injured people into four levels: priority 1 (red), priority 2 (yellow), priority 3 (green) and priority 4 (black). This preliminary version was presented in an international meeting [23] and because of the comments and opinions raised, it was decided to created a new code "violet" for suspected death and change priority 4 (black) to priority * (violet), since it is not considered ethical to remotely classify a person as dead, but should be triaged in situ when the first responders can access the scene of the accident. This Aerial Remote Triage System (ARTS) algorithm was tested through three small simulations [24].

## Phase 3. Evaluation of the algorithm by experts

The ARTS algorithm (Fig 1) was evaluated by a panel of experts through an online survey [25]. The experts were intentionally recruited by searching for health professionals who have published on the use of drones in healthcare or in health emergencies. The panel included professionals from different disciplines, different settings (academic and healthcare), and location (Spain and other countries). They were first contacted by e-mail or telephone, asking for verbal consent to participate. Fifteen experts accepted and the link for the survey was sent to them. The aim of the study and instructions to evaluate the ARTS was displayed at the beginning of the survey. Participants were informed that the data recorded were to be used in this study and asked to accept. The first round of the survey (algorithm evaluation) was run from 27 May to 21 June 2020, and the second round (simulation videos evaluation) from 15 to 28 February 2021. The questionnaires for experts are shown in S1 and S2 Appendices.

For the first round of the survey, the form included three sections. Section 1 had some questions about respondents´ education, work field and years of experience in emergencies and disasters. Section 2 had eleven statements about the use of drones in healthcare and section 3 had thirteen statements about the ARTS triage algorithm. The statements of these sections were written based on the information obtained from the literature review. The participants were asked to indicate their level of agreement with each statement on a four-point Likert scale from one (completely disagree) to four (completely agree); and also encouraged to add comments to any of the statements, especially when they were in complete or partial disagreement.

For the second round, the experts were asked to watch seven short videos showing footage taken with a drone camera in simulated emergencies for the different steps of the ARTS algorithm (scene size up, recruitment, bleeding assessment, manual compression, consciousness assessment, life sign assessment and recovery position) and to rate how useful the drone would be in each case (on a 5-points scale). Experts had the possibility to write their comments on the interactions seen in the videos.

Reminders were sent to improve the response rate. Data from the survey were tabulated for analysis and the name of the experts removed and replaced by a code. No published data identify individuals, institutions, or organizations. The IRB of the University of Jaen made the exemption from ethical approval for this study, because no personal data were recorded.

The agreement was estimated from the content validity ratio (CVR), calculated as $CVR = (n_e - N/2)/(N/2)$, where $n_e$ represents the number of experts that agreed with a statement (score of 3 or 4 in the 1st round and score of 4 or 5 in the 2nd) and N represents the number of experts [26]. The optimal CVR value is set to 0.49 for a 15 experts panel [27]. In addition, a qualitative synthesis of the experts comments was done.

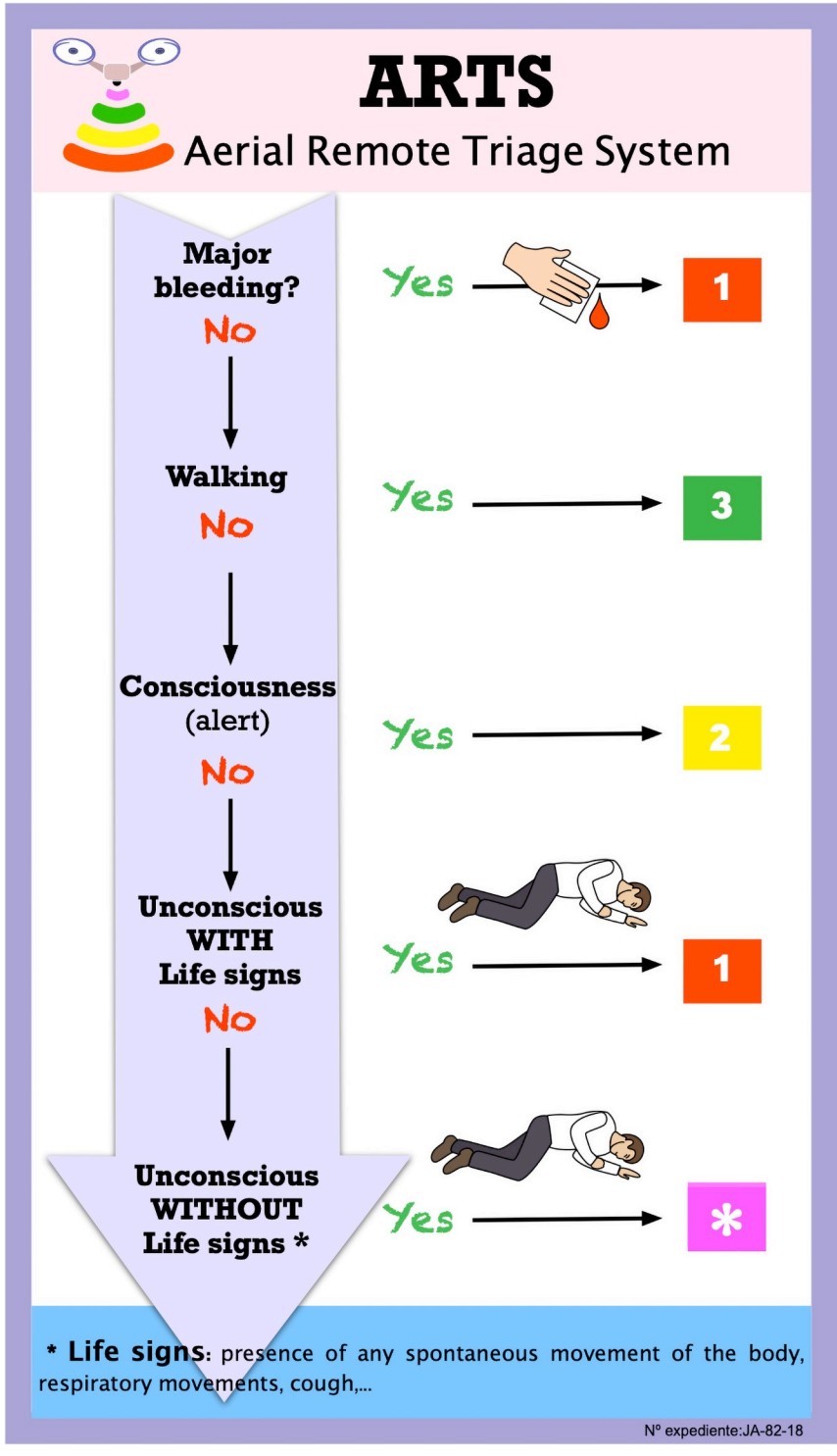

**Fig 1. Aerial Remote Triage System.**

**Table 1. Demographic characteristics of the experts (N = 15).**

| Characteristic | n | % |
|---|---|---|
| Degree | | |
| Medicine | 5 | 33.33% |
| Nursing | 7 | 46.66% |
| Healthcare emergency technician | 1 | 6.67% |
| Ph.D. epidemiologist | 1 | 6.67% |
| Critical care paramedic | 1 | 6.67% |
| Work field | | |
| Academic | 6 | 40% |
| Healthcare | 6 | 40% |
| Academic and assistance | 2 | 13.33% |
| Disaster, emergency, and military medicine | 1 | 6.67% |
| Years of professional experience, mean (SD) | 19.53 | 6.90 |

## Results

The demographic characteristics of the 15 experts who agreed to participate are described in Table 1. Both nursing and medical professionals were represented on the panel, and they had broad experience in the field of emergency.

### Content validation

The results for content validation obtained after the first round are shown in Tables 2 and 3. Table 2 shows the statements about the use of drones in emergencies and disasters, all but one of which reach the optimal level of agreement. Table 3 shows the statements about triage with the ARTS algorithm with four steps and four assigned color codes. Two statements did not reach the optimal level of agreement, one on the order for the remote assessment and the other on the use of the code VIOLET for an injured person who is suspected to be dead.

 Table 4 shows the results after the second round when several videos with simulated scenes of remote assessment with drones were presented to the experts. Two statements did not reach optimal agreement; one on providing instructions to bystanders using the drone loudspeakers, and the other on remotely identifying respiratory movements with the drone.

 The full dataset is available in Figshare [28].

### Qualitative evaluation and experts' comments

The experts made some comments and suggestions in the open box for each statement, which are summarized and displayed grouped by topic on Table 5.

### Use of drones in emergencies/disasters

The experts made some specific remarks about the 11 statements on the use of drones in health emergency situations.

 **Statements 1 and 2.** There was a high agreement on the usefulness of the use of drones in complex health emergencies or in areas of difficult access for an initial evaluation of the scenario. The experts highlighted the importance of gathering the information in a health emergency by means of aerial images when there is no way to access the site. Especially, in chemical, biological, radiological and nuclear (CBRN) emergencies, such as radiation risk in nuclear incidents, chemical spills or suicide attacks, drones can help by avoiding exposure of people and identifying the characteristics and magnitude of the incident.

**Table 2. Content evaluation of statements about the use of the drones in health emergencies.**

| | Statements | CVR | Mean | SD |
|---|---|---|---|---|
| 1 | The use of drones in complex health emergencies (nuclear, radiological, biological, chemical, terrorist attacks, active shooting, suicide bombs) or in hard-to-reach areas may be a valid option and preferable to physical access at the initial stage of the emergency. | 1 | 3.80 | 0.41 |
| 2 | The deployment of unmanned aircraft (drones) in complex health emergency scenarios (nuclear, radiological, biological, chemical, terrorist attacks, active shooting, suicide bombs) or without initial ground access, can help in the first assessment of the scenario (type of incident, added risks, number of injured people and their distribution in the place, access and evacuation routes). | 0.87 | 3.67 | 0.62 |
| 3 | The collection of relevant information for clinical assessment through cameras, sensors, and on-board systems in drones flying over complex health emergency scenarios may be considered as an alternative when ground access is not feasible. | 0.87 | 3.53 | 0.83 |
| 4 | The information provided by cameras, sensors, and on-board systems in drones flying over complex health emergency scenarios can be very useful in the decision-making process for managing the emergency. | 1 | 3.67 | 4.49 |
| 5 | A drone with speakers can provide remote medical support to injured people by using indications broadcast through the speakers. | 0.73 | 3.47 | 0.74 |
| 6 | A drone with speakers can communicate support messages to the injured people in complex health emergencies where access is impossible. | 0.87 | 3.60 | 0.63 |
| 7 | A drone with speakers can ask all persons involved in the emergency that are able to walk (bystanders) to remain standing, along with others who are not able to do so, in order to try to help them, if they consider themselves capable of helping. | 1 | 3.67 | 0.49 |
| 8 | A drone with speakers allows instructions for self-protections to be given to injured people. | 1 | 3.67 | 0.49 |
| 9 | A drone with speakers can be used to instruct a bystander to place injured people in the recovery position or to employ some hemostatic techniques to potentially exsanguinating wounds. | 0.6 | 3.40 | 0.83 |
| 10 | A drone can carry useful medical equipment to help in an emergency, such as medication (antidote auto-injectors, analgesia), bleeding kits, and isothermal sheet, before the arrival of the first responders to the place. | 0.87 | 3.60 | 0.63 |
| 11 | A drone deployed flying over a complex health emergency scenario allows individual assessment of injured people and healthcare prioritization through images and speakers. | 0.33* | 2.93 | 0.96 |

* Value under the optimal CVR value. CVR: Content validity ratio.

**Statement 3.** There was high agreement on the collection of relevant information for clinical assessment through drones when ground access is not feasible. Most experts mentioned the safety and speed of a drone in providing information on the injured people when access to the scenario is not possible, such as in a flood. In other cases, doubts have been expressed about being able to make this clinical assessment due to limitations of the device, the legal restrictions on the use of drones in some countries, or the difficulty of assessing the condition of the injured people without being in situ.

**Statement 4.** There was high agreement on the usefulness of the information provided by a drone in the decision-making process for emergency management. Experts supported the idea that drones could be very useful in the management of a health emergency because they can provide information about potential hazards (such as the presence of dangerous gases or chemicals) or an estimation of the number of injured people in the scenario and their location. As one of the experts said "More data and more information is always better".

**Statement 5.** There was moderate agreement on the use of drone´s loudspeakers to provide remote medical support to injured people. Some experts pointed out that it is possible to give instructions for compression of exsanguinating bleeding, to perform basic life support or to place an unconscious person in the recovery position. However, other experts were concerned that the communication is only in one-way because the drones do not allow the injured

**Table 3. Content evaluation of statements about triage with ARTS algorithm.**

| | Statements | CVR | Mean | SD |
|---|---|---|---|---|
| 1 | The first step before starting the triage will be to broadcast an audio message through the drone speakers instructing people who can walk to remain standing and those who cannot walk to make some movement. | 0.83 | 3.75 | 0.62 |
| 2 | The assessment of healthcare priority will start with those injured people who do not respond to messages from the drone speakers. | 0.83 | 3.67 | 0.65 |
| 3 | In the assessment of healthcare priority supported by drones, the aspects to be evaluated would be, in this order: 1°, major bleeding, 2°; walking, 3°; consciousness (alertness); 4°, signs of life. | 0.17* | 2.92 | 1.08 |
| 4 | In the assessment of healthcare priority supported by drones, if the injured person presents wounds with important bleeding (exsanguinating according to criteria of Hartford Consensus), priority 1 (RED) will be assigned. | 0.5 | 3.25 | 1.06 |
| 5 | In the assessment of healthcare priority supported by drones, if the injured person presents wounds with important bleeding (exsanguinating according to criteria of Hartford Consensus), the employment of some hemostatic techniques will be instructed to the injured person or some bystander through the drone speakers, if possible. | 1 | 3.83 | 0.39 |
| 6 | In the assessment of healthcare priority supported by drones, if the injured person can walk, priority 3 (GREEN) will be assigned. | 0.67 | 3.58 | 1 |
| 7 | In the assessment of healthcare priority supported by drones, if the injured person does not walk, but he or she is conscious (responds to audio messages), priority 2 (YELLOW) will be assigned. | 0.67 | 3.25 | 0.97 |
| 8 | In the assessment of healthcare priority supported by drones, if the injured person is apparently unconscious but shows signs of life (spontaneous movements of the body, respiratory movements, coughing. . .), priority 1 (RED) will be assigned. | 0.83 | 3.67 | 0.65 |
| 9 | In the assessment of healthcare priority supported by drones, if the injured person is apparently unconscious and with signs of life, bystanders will be instructed through drone speakers to place the injured person in the recovery position, if possible. | 1 | 3.67 | 0.49 |
| 10 | In the assessment of healthcare priority supported by drones, if the injured person is apparently unconscious and without signs of life (spontaneous movements of the body, respiratory movements, coughing. . .), priority * (VIOLET) will be assigned. | 0.33* | 3.08 | 0.90 |
| 11 | In the assessment of healthcare priority supported by drones, the PRIORITY * (VIOLET) indicates that the injured person is waiting for in situ re-assessment by the first responders. | 0.83 | 3.58 | 0.67 |
| 12 | In the assessment of healthcare priority supported by drones, if no signs of life can be observed in the injured person (spontaneous movements of the body, respiratory movements, coughing. . .), bystanders will be instructed through drone speakers to place the injured person in the recovery position, if possible. | 0.67 | 3.42 | 0.79 |
| 13 | Providing the first responders with the results of the Aerial Remote Triage System before they have access to the site, can be useful and helpful. | 0.83 | 3.58 | 0.67 |

* Value under the optimal CVR value. CVR: Content validity ratio.

people to communicate with the rescuers; so it was proposed the transport of walkie-talkies or mobile phones. Another problem mentioned was that the noisy environment in typical collective emergency scenarios could produce audio interferences.

**Statements 6.** There was high agreement on the usefulness of the emotional support provided by loudspeakers on drones. Experts agreed that it is possible and beneficial, despite the unidirectional nature of the communication. One of the experts proposed incorporating a screen in drone that shows the image of the person giving the instructions to make the process more humane.

**Statements 7 and 8.** There was high agreement on the capacity to ask the bystanders to help other injured people through messages broadcast by the drone's loudspeaker. One of the comments shows the value of using the drone to instruct the injured people to move to safer areas. Experts also mentioned some problems, the possibility that the confusion generated in a

**Table 4. Content evaluation of statements about image assessment.**

| | Statement | CVR | Mean | SD |
|---|---|---|---|---|
| 1 | After watching the following video (scene size up), how useful is a drone for the initial assessment of a scenario in an emergency. | 0.80 | 3.58 | 1.25 |
| 2 | After watching the following video (recruitment), how likely do you think it is that bystanders could be recruited in an emergency situation using an on-board loudspeaker system on a drone. | 0.60 | 3.50 | 1.03 |
| 3 | After watching the following video (bleeding assessment), how useful a drone is in assessing an exsanguinating wound. | 0.60 | 3.25 | 0.88 |
| 4 | After watching the following video (manual compression), how likely do you think it is that the injured person/bystander will follow the instructions broadcast over a drone-borne loudspeaker system to manually compress the wound. | 0.40* | 3.08 | 0.82 |
| 5 | After watching the following video (consciousness assessment), how useful a drone is to identify signs of life, such as voluntary movements in a conscious injured person. | 0.80 | 3.50 | 0.63 |
| 6 | After watching the following video (life signs assessment), how useful a drone is in identifying signs of life such as respiratory movements in an unconscious injured person. | 0* | 2.92 | 1.35 |
| 7 | After watching the following video (recovery position), how likely do you think it is that the bystander will follow the instructions for positioning the injured person in the recovery position. | 0.60 | 3.42 | 0.74 |

* Value under the optimal CVR value. CVR: Content validity ratio.

MCI and the stress secondary to the impact, may hinder following orders; and some doubts as to whether a message issued by a device can generate enough trust to follow it.

**Statement 9.** There was moderate agreement on the possibility of giving specific instructions to the bystanders to perform specific manoeuvres (such as placing injured people in the recovery position or even applying hemostasis techniques to wounds). Some experts were reluctant about the capability of the bystanders to apply these life-saving interventions.

**Statement 10.** There was high agreement on the possibility of delivering small medical equipment by drone, before the arrival of the first responders. This equipment could be medication (antidote auto-injectors, analgesia), bleeding control kit or isothermal sheets. The experts also proposed other materials or devices, such as glucagon kit, automatic external defibrillators, or even food.

**Statement 11.** There was poor agreement on the possibility to perform a remote individual assessment of injured people using a drone. Experts mentioned the difficulty of individualized assessment of injured people in some cases (eg. when lying in a prone position), or the difficulty of assessing physiological parameters remotely. In this way, experts urge that triage be made as simple as possible.

## Triage with the ARTS algorithm

Following, some comments expressed by the experts on the 13 statements describing the remote triage with the ARTS algorithm.

**Statement 1.** There was high agreement to broadcast a message over the drone's loudspeaker asking people who can walk stand up or those who cannot walk make some movement. However, some experts mentioned the difficulty of hearing and understanding the message in noisy scenarios.

**Statement 2.** There was high agreement that priority should be given to injured people who do not respond to the drone's message. Only one expert mentions that perhaps those injured people who do not respond are the least likely to survive and if a remote triage has been chosen the time until the attention may be long, while other injured people are neglected.

**Table 5. Experts´ comments regarding the use of drones in health emergencies grouped by topic.**

| Use of drones in health emergencies | | |
|---|---|---|
| 1. Use of drones in complex health emergency scenarios with no physical access | • For the evaluation of the incident<br>• For better management of the emergency | "The information collected . . . is relevant."<br>". . .to introduce a drone to obtain images,. . . is vital."<br>"Whenever there is a foreseeable risk on the scene, we should opt for drones or robots that allow us to explore the situation without risk to human lives"<br>". . .the drones can provide information about the victims in a safe and agile manner.. . . |
| 2. Usefulness of instructions broadcast through the drone's loudspeakers | • To provide medical support<br>• To communicate expressions of support to injured people<br>• To request the collaboration of bystanderss in the realization of life-saving interventions<br>• To indicate measures of self-protection to the injured person | "A drone can provide emotional support and inform victims that emergency services are approaching and looking out for them."<br>"It can be an excellent support alternative to the traditional one, for example, by indicating how to apply a tourniquet, or how to move a victim in a safe side position or to perform basic life-saving interventions."<br>"Yes, you can help by giving instructions to a person who may not have any information about where they are." |
| 3. Usefulness of drone for the carriage of medical material | | "Yes, it could carry insulin in pens for diabetics, for example, or glucagon, or even some antidote for poisoning or chemical accidents, knowing for sure the agent producing the poisoning." |
| 4. Usefulness in assessing injured people and prioritizing their care | | "Yes, having microphones and emergency personnel managing the incident from the transmitter side of the loudspeaker would be a perfectly valid and useful procedure." |
| Aerial Remote Triage System | | |
| 1. Previous step, broadcast a message (audio) through the drone | • To identify injured people who can walk or move<br>• To begin prioritizing unresponsive injured people | "In the message, I would first add that those who understand the message should make a gesture (e.g., raise a hand)."<br>"I would do it this way, to assess whether they are conscious or present exitus, or in life-threatening difficulties, are the red, black or grey victims, and we must differentiate them in this way. I think it's appropriate to do it this way." |
| 2. Sequence of aspects to be evaluated: 1˚, major bleeding; 2˚, walking; 3˚, consciousness (alert), 4˚, signs of life. | • To assign priority in the remote evaluation process | "This is the pattern of action according to the triage START."<br>"Those who need life-saving interventions first so they could be addressed." |
| 3. Injured people with bleeding injury (Hartford Consensus) | • Priority 1 (RED).<br>• The drone loudspeaker indication of haemostasis techniques | "I think it's clear, given the mortality associated with a victim with significant bleeding that is exsanguinating."<br>"Right. Ideally. . . the drone should have options to release a tourniquet and indicate its placement."<br>"Direct compression is easy enough to be performed by laypersons." |
| 4. Injured people with walking ability | • Priority 3 (GREEN) | "This is the pattern followed by different triage systems START, SHORT, Sieve, MRCC. . ." |
| 5. Injured people with no walking ability but conscious | • Priority 2 (YELLOW) | ". . .yellow in revaluation. The bad thing is that we have no option to easily assess the neurological one since it can be a yellow that is red in fact because of ECT or with neurological deficits." |
| 6. Unconscious injured people with signs of life (presence of any spontaneous body movement, breathing movements, coughing. . .) | • Priority 1 (RED)<br>• Recovery position | "Yes, I agree, it's the data that indicate us he's at least breathing and has signs of life."<br>"This would be ideal."<br>"We must be very careful here as the victim may have injuries. . ." |
| 7. Injured people apparently unconscious with no signs of life | • Priority * (VIOLET)<br>• Injured people waiting to be reassessed in situ by the first responders<br>• Recovery position | "Yes, I am interested in reconfirming the victim's condition in person, for confirmation of the expected diagnosis."<br>"I agree when violet is an unevaluated patient."<br>"As long as it does not compromise the bystander in becoming another victim, I see fit." |
| 8. Communication to the first responders of the findings of the Aerial Remote Triage System | • The usefulness of the information collected through the drones | "Yes, because it speeds up the organization of the team and the discrimination of attention to the victim who needs it first.<br>"It can be of great help."<br>"Information is the most important thing in a catastrophe. . . To my mind, that would not be the best thing. . . the best thing is for the drone to position the victim with its triage and take a picture." |

**Statement 3.** There was a poor agreement on the sequence of assessment and prioritization for triage (1st major bleeding, 2nd walking, 3rd consciousness, 4th signs of life). Some experts supported that this order is followed in other triage systems such as START, but others considered that it is difficult to assess bleeding using images or that it could be better to assess first if the injured people can walk, so they can move to a safe place.

**Statement 4.** There was moderate agreement on assigning priority 1 (red) to an injured person with significant bleeding, but the difficulty of assessing bleeding remotely was mentioned.

**Statement 5.** There was a high agreement on the importance of applying hemostatic techniques to the injured people with bleeding. The possibility of sending tourniquets with drone was introduced but the placement of these requires some training, so it was determined that the most appropriate technique to use by a bystander or the self-application by the injured person would be direct compression.

**Statement 6.** There was a moderate agreement on remotely assigning priority 3 (green) to injured people that can walk. However, there were certain nuances about this category; the fact that they can walk does not indicate the absence of injuries. Hence, the recommendation for continuous re-assessment of these people once they have been labelled green.

**Statement 7.** There was a moderate agreement on assigning priority 2 (yellow) to injured people that are conscious but cannot walk. Some experts expressed concerns about the limitations of the drone to accurately identify these people, being important to discern if a yellow, green or a red priority should be applied. According to some experts, through a drone you might suspect a category but not rule it out. Others comment on the limitations of an aerial triage compared to standard triage, since with the former you can only count, evaluate and send messages.

**Statement 8.** There was a high agreement on assigning priority 1 (red) to apparently unconscious injured people but with signs of life. Some experts commented on the difficulty to detect, through the images provided by the drone, the presence of signs of life, such as respiratory movements. For others, that is simply a difficult challenge to achieve.

**Statement 9.** There was a high agreement on sending instructions to bystanders to put injured people in the recovery position. However, some experts commented that if a spinal injury is suspected, so this intervention should be avoided and the injured person should not be mobilized.

**Statement 10.** There was a poor agreement on assigning the code violet for injured people without signs of life. Experts commented that there are some standardized triage codes for this situation, such as black, red or blue.

**Statement 11.** There was a high agreement on the need to re-assessment in situ for injured people coded violet or without signs of life.

**Statement 12.** There was a moderate agreement on the recommendation to put the injured people without signs of life in the recovery position. Some experts mentioned the need to be cautious in conducting such interventions in the face of the possibility that the injured person has spinal injuries or injuries incompatible with life.

**Statement 13.** There was a high agreement on the usefulness in providing to the first responders in situ with the data obtained by the remote triage with ARTS. They also highlighted the additional advantage of determining the position of each of the injured people labelled with priorities, so the attention and use of resources would be faster and more efficient.

## Simulated scenario evaluation

After seeing the videos of simulated scenarios applying the ARTS algorithm filmed by a drone, the experts considered as useful the overall assessment of the scenario (scene, number of

injured people, risks, need for firefighters), especially for shortening response times. In addition, they considered that voice prompts bring peace of mind to the scene, because that makes it possible to send to the injured people the message that qualified personnel are watching and helping them. The main problem for some experts was the request for collaboration to bystanders or the injured people themselves, since this implies a human factor and not only the technical capacity to transmit messages to the injured people.

Four aspects did not agree among the experts: the assessment of breathing, the identification of major hemorrhages, the application of direct compression to bleeding wounds and the recommendation to place the injured people in the recovery position. Several external factors in the scenario such as weather, lighting (natural/artificial light), the type of pavement, type of clothing of the injured, or their position/location are all involved and might alter the accuracy of the remote inspection. In addition, one expert mentioned that manual compression in large hemorrhages would be insufficient and could provoke nervousness in the person applying it when observing that the bleeding does not stop; considering a tourniquet sent by the drone as a better option.

## Discussion

This article reports on the development and initial validation of a system for remote triage by drones that may be useful in MCIs. The Aerial Remote Triage System is an algorithm structured in five steps that establish four levels of priority (labelled as red, yellow, green and violet). ARTS is proposed as an alternative triage system to use in MCI scenarios in which (or while) it is not possible to use conventional land-based triage; but it does not intend to replace the current triage systems when the place of the MCI can be reached. Overall, the experts of the panel surveyed supported the concept of remote triage by drones and the steps of the ARTS algorithm, but some specific considerations emerged.

### Assessment of MCIs by drones

Regarding the use of drones for initial assessment and intervention in MCI, three points arose from the survey with the experts: remote communication, intervention with the bystanders and individual assessment of the injured people.

Previous studies have used voice commands to communicate with injured people in a MCI [21,29–31]. Holgerson et al. [31] showed some strategies to overcome communication difficulties at the scene of a MCI, such as the reservation of priority lines in the cell phone network and the use of loudspeakers to inform the crowds. Jain et al. [4] reported that, in simulated MCI scenarios, injured people were able to follow the instructions given by the first responders through a loudspeaker incorporated in a drone; also other study by Sanfridsson et al. [32] found that bystanders perceived easier the interactions with the drone loudspeaker than with their own mobile phone in speakerphone mode. Carter et al. [33] analyzed the application of collective psychology to develop recommendations for the management of mass decontamination CBRN incidents. They concluded that the communication strategies perceived as most effective were those that focused on health, information about the actions that responders were taking, and practical information. The injured people could initiate the actions for self-decontamination as delivered through loudspeakers by the first responders.

The Hartford Consensus [34] defined bystanders as people who witnessed the incident but maintains his or her ability to help others who are injured. The ARTS includes to give some basic instructions to bystanders to perform some life-saving manoeuvres when appropriate (compression of bleeding injuries, placing the injured people on the recovery position or using automated external defibrillators) [32,34]. In contrast with the traditional land-based triage systems, the ARTS recommend not to ask the injured people to leave the area of the incident

(if this area is safe enough) so the bystanders could help them. Previous studies reported on the usefulness of life-saving interventions in MCIs, such as control of exsanguinating hemorrhage, the recovery position [35,36] and the use of antidote-type medication with self-injector [20]. There is controversy over the recovery position for unconscious people; guidelines describe this intervention as useful in all unconscious injured people with spontaneous breathing [37,38], but recommend caution in injured people when spinal trauma is suspected. A study that compared the recovery position with the HAINES (High Arm IN Endangered Spine) technique [39] on cadavers did not reach evidence on which is the best version of the recovery position in injured people with cervical injuries [40].

The ARTS includes the possibility of performing an individual assessment of the injured people looking for signs of life, using a drone equipped with a camera; however this is a point without consensus among the experts. Some previous studies support this possibility, it was reported that 82% of health professionals involved in triage in a MCI adequately classified 12 of 15 injured using the SALT method [16] and that a drone could identify the number of people involved in an MCI, their general condition, the presence of respiratory movements and the state of consciousness after simulations in different types of accidents [7]. Regarding the difficulty to establish some physiological parameters when injured people are in prone position, a recent study reports that it is possible to determine whether a person is alive or dead from the processing of images captured by a drone, after the evaluation of the cardiorespiratory movements of injured people in the supine, prone or lateral position [41]. The ARTS is a simple remote triage system, which does not require special skills to determine physiological parameters and is adapted for remote use, which might prevent errors such as those made when using the SALT method for remote triage [16].

## Remote triage using the ARTS

The first step in triage with ARTS is to identify the injured people who can walk or move by emitting a message through the drone's loudspeaker. That is a usual process in different triage systems and included in the Hartford-TECC Compendium [33]. The study by Jain et al. [4] showed that a drone is an equally effective alternative to a team of rescuers for an initial classification of injured people by issuing a message to evacuate those who are able to walk. Thus, once the less seriously injured people have been detected, it is possible to begin the assessment of the injured people who have not responded to the message and may be more likely to benefit from life-saving interventions applied by bystanders [35]. The sequence of assessment proposed in the ARTS (1-mayor bleeding, 2-walking, 3-consciousness, 4-signs of life) did not reach a full consensus among the panel experts; however this sequence is similar to that proposed by several triage guides [33,42] or in a basic triage system for the lay public [43].

Early identification of injured people with major bleeding is an important step in the ARTS. Although some experts considered that the injured people should first be assessed whether or not they can walk to classify them as green; one of the basic principles of triage is fairness, therefore it is considered preferable to assist first the more seriously injured people [19]. A person with major hemorrhage can go into shock and die within minutes [44], so interventions such as direct wound compression could save his or her life [45,46]. In addition, a drone may deliver some devices (tourniquets, dressing or painkillers) [47], when direct wound compression is not sufficient. Nevertheless, the difficulty of remote assessment and quantification of the hemorrhage is a challenge. A study investigated the ability of laypeople to visually assess blood loss by watching short videos of individuals suffering a hemorrhage. It revealed that laypeople overestimated small volumes of blood loss (50–200 ml) and underestimated larger volumes (400 –1900 ml) [48]. A similar situation happened with clinicians, regardless of

their training and skills [49]. Thus, the visual estimation of blood loss is usually inaccurate; therefore, it is an inherent limitation of all triage systems.

## Prioritization codes in ARTS

The use of color codes for triage is generally accepted as a way of indicating the life expectancy of the injured people, guiding the urgency of the intervention and the amount of therapeutic effort to be made [35]. The most frequent color codes used are red, yellow, green and black. However, other colors have been proposed for specific situations: white for injured people with pulse but do not breath (more severe injuries than people coded as red, but that cannot be coded as black because they have signs of life) in Homebush triage [50] or the orange in the Triage Early Warning Score model [19]. Within the triage system that we are developing (ARTS) the color code violet is proposed for injured people without apparent signs of life, instead of the color code black, because it is not possible to remotely confirm the death. The use of this color indicates, on the one hand, the extreme seriousness of the injured person awaiting confirmation of his condition by the first responders when they arrive at the scene and, on the other hand, as a way of differentiating it from the red coded.

Color code green identifies to injured people with priority 3 (can wait). However, it is important to take into account the need to re-evaluate and monitor them continuously once they have been identified, according to most of the traditional triage systems [20,35,51]. Color code yellow identifies injured people with priority 2 (observation). In the ARTS, given the difficulty to remotely assess respiration, it is proposed to assume that any conscious person is breathing and has a heartbeat enough to maintain cerebral perfusion, as stated by other systems [51]. In our study, some experts expressed doubts about the possibilities of identifying this type of injured people by the drone, because breathing cannot be assessed. Recent developments in technology have shown that both the heart and respiratory rate may be measured, from images provided by a drone, through photoplethysmography and movement magnification techniques [52,53]. So, the technology already exists, although more research and development are needed to implement in the MCIs evaluation by drones. Color code red is used to identify injured people with priority 1 (immediate attention) those seriously injured but with signs of life and a chance of survival. In the ARTS the presence of external respiratory movements, spontaneous body movements or cough are taken as signs of life [37]. In our survey, some experts commented about the difficulty to detect signs of life by drone's camera. We agree that this may be a challenge sometimes, but recent advances are paving the way, such as new technologies in drones that allow to remotely assess vital signs through different imaging techniques [52] or the use of two-way audio system, able to capture voices and determine their location in land, for detecting injured people under the rumble [54]. A recent study investigated the ability of small drones to evaluate breathing both after landing on the bodies of injured people or when hovering over them without landing [55]. This study reported that, when the drone landed, breathing was correctly determined in 100% of participants in the supine position and in 96% in the prone position; and when the drone just was kept hovering over the people, breathing status was misinterpreted in 29.6% of the participants lying in the supine position and in 36.1% in the prone position.

This study has some limitations that should be taken into account. That is an initial validation of the concept of triage through drones and the ARTS algorithm. Its development is based on a literature search and extensive practical expertise in the field of health emergencies of some authors, combined with the contributions of the expert panel to enrich and refine the concepts. However, that is mostly a theoretical model that has to be tested, first in simulated scenarios and then in real MCIs. There are some specific issues in the ARTS that generated

doubts from the experts and should be tested using a drone for triage: the possibility to identify major bleedings, to assess whether the injured people breath and to find and ask bystanders to help the injured people.

With the growing use of drones in health emergencies and disasters, it is necessary to develop regulations for their correct use [56] and to balance risks and benefits in each situation [5,7,57]. Among the regulations adopted in some countries, we can find the necessity to obtain a certificate to pilot the drone or the use of drones only during daylight hours, with the drone in the line of sight and in controlled airspace [58]. It would also be necessary to control intrinsic factors of the drone such as flight speed, payload, characteristics and sensors that a medicalized drone must possess; as well as extrinsic factors, such as the weather conditions in which it can fly [59–62]. Currently, the main priority is to conduct more research using drones in real situations since most of them have been performed in simulated situations. In addition, the Spanish Society of Emergency Medicine has proposed the creation of alternative triage cards like virtual triage cards based on images provided by a drone, so that an image could be used to locate where the injured people are and their priority of care [13].

## Conclusions

The ARTS is an algorithm for remote triage by drones in MCIs that may be useful for an initial and fast assessment of scenarios of complex health emergencies with difficult or delayed access, due to physical barriers or CBRN situations. The use of a drone can speed up the process of obtaining information about the emergency. The ARTS includes four steps for the assessment: first, major bleeding; second, walking; third, consciousness; and fourth, signs of life; and classifies the injured people into priority four categories: priority 1 (red), priority 2 (yellow), priority 3 (green), and priority * (violet). The system includes the possibility to indicate life-saving interventions to injured people and bystanders, such as the compression of bleeding injuries or the recovery position.

## Supporting information

**S1 Appendix. Questionnaire for experts: Algorithm assessment.**
(DOCX)

**S2 Appendix. Questionnaire for experts: Images assessment.**
(DOCX)

## Acknowledgments

The authors want to thank the experts who participated in the surveys in the process of developing the ARTS.

## Author Contributions

**Conceptualization:** Sixto Cámara-Anguita, José María López-Hens.

**Data curation:** Cristina Álvarez-García, Sixto Cámara-Anguita, José María López-Hens, Nani Granero-Moya, María Dolores López-Franco, Inés María-Comino-Sanz, Sebastián Sanz-Martos, Pedro Luis Pancorbo-Hidalgo.

**Formal analysis:** Cristina Álvarez-García, Sixto Cámara-Anguita, José María López-Hens, Nani Granero-Moya, María Dolores López-Franco, Inés María-Comino-Sanz, Sebastián Sanz-Martos, Pedro Luis Pancorbo-Hidalgo.

**Investigation:** Cristina Álvarez-García, Sixto Cámara-Anguita, José María López-Hens, Nani Granero-Moya, María Dolores López-Franco, Inés María-Comino-Sanz, Sebastián Sanz-Martos, Pedro Luis Pancorbo-Hidalgo.

**Methodology:** Cristina Álvarez-García, Sixto Cámara-Anguita, José María López-Hens, Nani Granero-Moya, María Dolores López-Franco, Inés María-Comino-Sanz, Sebastián Sanz-Martos, Pedro Luis Pancorbo-Hidalgo.

**Supervision:** Cristina Álvarez-García, Pedro Luis Pancorbo-Hidalgo.

**Validation:** Cristina Álvarez-García.

**Visualization:** Cristina Álvarez-García, Sixto Cámara-Anguita, José María López-Hens, Nani Granero-Moya, María Dolores López-Franco, Inés María-Comino-Sanz, Sebastián Sanz-Martos, Pedro Luis Pancorbo-Hidalgo.

**Writing – original draft:** Cristina Álvarez-García, Sixto Cámara-Anguita, José María López-Hens, Nani Granero-Moya, María Dolores López-Franco, Inés María-Comino-Sanz, Sebastián Sanz-Martos, Pedro Luis Pancorbo-Hidalgo.

**Writing – review & editing:** Cristina Álvarez-García, Sixto Cámara-Anguita, José María López-Hens, Nani Granero-Moya, María Dolores López-Franco, Inés María-Comino-Sanz, Sebastián Sanz-Martos, Pedro Luis Pancorbo-Hidalgo.

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
