## [Decision Letter · Decision Letter 0]

5 Jan 2021

PONE-D-20-36599

Development of the Aerial Remote Triage System: Result of a survey of international experts

PLOS ONE

Dear Dr. Lopez-Franco,

Thank you for submitting your manuscript to PLOS ONE. After careful consideration, we feel that it has merit but does not fully meet PLOS ONE’s publication criteria as it currently stands. Therefore, we invite you to submit a revised version of the manuscript that addresses the points raised during the review process.

It is an interesting study. However, there are some important shortcomings. Please, consider the proposal of reviewer 1: It would be greatly appreciated if the authors would film at least a small number of actors with a drone as a scenario, so that experts may make their decisions. Such an additional part should be carried out and incorporated into the work. I hope you are willing to do so.

We look forward to receiving your revised manuscript.

Kind regards,

Hans-Peter Simmen, M.D., Professor of Surgery

Academic Editor

PLOS ONE

Journal Requirements:

3. Please provide additional details regarding participant consent.

In the ethics statement in the Methods and online submission information, please ensure that you have specified (i) whether consent was informed and (ii) what type you obtained (for instance, written or verbal, and if verbal, how it was documented and witnessed). If the need for consent was waived by the ethics committee, please include this information.”

4. In your Methods section, please provide additional information about the participant recruitment method and the demographic details of your participants. Please ensure you have provided sufficient details to replicate the analyses such as:

a) a description of any inclusion/exclusion criteria that were applied to participant recruitment,

b) descriptions of where participants were recruited and where the research took place.

5. Please include additional information regarding the survey or questionnaire used in the study and ensure that you have provided sufficient details that others could replicate the analyses.

For instance, if you developed a questionnaire as part of this study and it is not under a copyright more restrictive than CC-BY, please include a copy, in both the original language and English, as Supporting Information.

6. Please note that according to our submission guidelines (http://journals.plos.org/plosone/s/submission-guidelines), outmoded terms and potentially stigmatizing labels should be changed to more current, acceptable terminology. For example: “victims” should be changed to “patients”.

7. To comply with PLOS ONE submission guidelines, in your Methods section, please provide additional information regarding your statistical analyses. For more information on PLOS ONE's expectations for statistical reporting, please see https://journals.plos.org/plosone/s/submission-guidelines.#loc-statistical-reporting.

8. We note that you have indicated that data from this study are available upon request. PLOS only allows data to be available upon request if there are legal or ethical restrictions on sharing data publicly. For information on unacceptable data access restrictions, please see http://journals.plos.org/plosone/s/data-availability#loc-unacceptable-data-access-restrictions.

9. Thank you for stating the following in the Competing Interests section:

'SCA AND JMLH are founders of Emergency Medical Drone cooperative'

Reviewers' comments:

Reviewer's Responses to Questions

**Comments to the Author**

1. Is the manuscript technically sound, and do the data support the conclusions?

Reviewer #1: Partly

Reviewer #2: Yes

2. Has the statistical analysis been performed appropriately and rigorously? 

Reviewer #1: No

Reviewer #2: Yes

3. Have the authors made all data underlying the findings in their manuscript fully available?

Reviewer #1: Yes

Reviewer #2: Yes

4. Is the manuscript presented in an intelligible fashion and written in standard English?

Reviewer #1: Yes

Reviewer #2: Yes

5. Review Comments to the Author

Reviewer #1: This investigation is concerned with an important point in developing a system for dealing with a mass casualty. An expert survey should clarify whether it makes sense to carry out a triage on the use of drones.

The data analysis used includes the essential databases, but too few search terms are used. The search terms terrorist attack, earthquake, poison attack and the like should also be used here.

It is also unclear how the questionnaire was created. For this purpose, the material and methods must be used to specify how the questionnaire was created.

In the end, only a survey of experts was carried out, the results were clearly summarized in a table. So it is ultimately a piece of literature and an expert survey. a statistical scientific evaluation was not carried out. In particular, no experimental study was carried out. Here one could have imagined showing the experts various pictures with injury patterns (actors, filmed by the drone) in order to determine to what extent the proposed triage is sensible and possible. such an experimental part should be reworked.

Reviewer #2: Thanks for this interesting paper. Its a step into the right direction to get a better overview in some mass-casualty incidents! I was amazed that the experts did not agree in the sequence of assignment of priority, for me, it is a simple a clear sequence. And as you wrote already, the triage should be mad as simple as possible.

The expert point of making the process more human in broadcasting the image is not a high priority in this situation, the additionally weight should be better used, for example to transport necessary equipment to the scene.

There is always a discussion about spinal injury and mobilization of a patient (statement 9).... But if you don't put hin in recovery position, he will be dead (maybe with or without paralysis, but he well be dead). -> If in doubt, put in recovery position ("life before limb").

6. PLOS authors have the option to publish the peer review history of their article (what does this mean?). If published, this will include your full peer review and any attached files.

Reviewer #1: No

Reviewer #2: No

---

## [Author Response · Author response to Decision Letter 0]

20 Apr 2021

The respond to reviewers is attached in a file named "Response to reviewers"

---

## [Editor Report · Decision Letter 1]

26 Apr 2021

Development of the Aerial Remote Triage System using drones in mass casualty scenarios:  a survey of international experts

PONE-D-20-36599R1

Dear Dr. Lopez-Franco,

We’re pleased to inform you that your manuscript has been judged scientifically suitable for publication and will be formally accepted for publication once it meets all outstanding technical requirements.

Kind regards,

Hans-Peter Simmen, M.D., Professor of Surgery

Academic Editor

PLOS ONE
---

## [Editor Report · Acceptance letter]

30 Apr 2021

PONE-D-20-36599R1 

Development of the Aerial Remote Triage System using drones in mass casualty scenarios:  A survey of international experts 

Dear Dr. López-Franco:

I'm pleased to inform you that your manuscript has been deemed suitable for publication in PLOS ONE. Congratulations! Your manuscript is now with our production department. 

Kind regards, 

on behalf of

Dr. Hans-Peter Simmen 

Academic Editor

PLOS ONE